# THE LIMITS OF FAIRNESS GAINS UNDER SCALING IN VISION MODELS

## ABSTRACT

Recent advances in computer vision indicate that increasing dataset size and model parameters substantially enhance model performance. Scaling laws derived from these observations provide valuable guidance for the design and optimization of large vision models. However, the impact of scaling on fairness within these models has yet to be systematically investigated. Here we empirically show that scaling model parameters and dataset size can improve fairness for certain protected attributes in downstream tasks. Our results demonstrate that, when using a loss function that jointly optimizes for utility and fairness, there exists a critical threshold in scaling beyond which fairness gains plateau. While scaling enhances fairness for some attributes, it does not eliminate disparities. These results emphasize that fairness in vision models requires more than scaling. Fairness techniques must be incorporated early in model development to address structural disparities and improve outcomes for all groups. This is especially crucial in sensitive domains such as medical imaging, where achieving equal representation and unbiased performance across diverse populations is essential for ethical and effective deployment.

## 1 INTRODUCTION

Self-supervised learning enables the utilization of large unlabeled datasets, thereby reducing reliance on costly and time-consuming manual annotation, particularly in domains such as medical imaging where expert-labeled data are scarce Azizi et al. (2023); Dippel et al. (2024); Zhou et al. (2023); Tu et al. (2023). By employing pre-training tasks, models acquire meaningful representations that can be efficiently adapted to a variety of downstream applications Chen et al. (2020); He et al. (2021); Darcet et al. (2025); Siméoni et al. (2025); Garrido et al. (2024). Although self-supervised approaches generally learn more robust and equitable model performance across imbalanced data then supervised learning Goyal et al. (2022), persistent biases in representation may still perpetuate discrimination, necessitating ongoing attention to fairness and generalizability Glocker et al. (2023); Queiroz et al. (2025b).

Despite advances in large-scale datasets and modern neural architectures, model bias remains pervasive, primarily because models learn and amplify underlying biases intrinsic to their training data Liu & He (2024); Zeng et al. (2024); Meister et al. (2023). In high-stakes domains such as medical imaging, these biases can systematically privilege certain groups while disadvantaging others, exacerbating health disparities and undermining equitable care Yang et al. (2024); Zhao & Gordon (2022). Consequently, developing rigorous fairness-aware models and representational techniques is essential for fostering a more equitable and democratic artificial intelligence landscape Bommasani et al. (2022); Queiroz et al. (2025b); Longpre et al. (2024).

Scaling laws establish power-law relationships between model performance and key design factors, notably data volume and model architecture, across domains such as vision and text Hernandez et al. (2021); Hestness et al. (2017); Kaplan et al. (2020); Zhai et al. (2022). Recent research investigates the link between bias and scaling laws in text domain, revealing that model scale and pretraining data influence social biases: larger models trained on internet data exhibit increased toxicity, whereas those trained on curated sources display stronger stereotypes, although downstream biases typically decrease with scale the model parameters Ali et al. (2024). In downstream tasks, finetuning dataset

size distribution alignment between the pretraining and downstream data significantly influence the scaling behavior Isik et al. (2025).

Fairness is critical in medical imaging, as model biases can exacerbate disparities across protected groups defined by age and gender Chen et al. (2023); Ricci Lara et al. (2022). This study systematically investigates how scaling both dataset size and model parameters in downstream tasks affects the loss and disparity in AUROC among protected and non protected attributes, revealing that neither increased data nor model parameters alone is sufficient to mitigate bias. Notably, the impact of scaling is contingent on the specific protected attributes and the domain of the dataset. Incorporating a fairness-targeted loss function exposes a threshold beyond which further increases in data or parameters yield negligible gains in bias mitigation. Consequently, these findings emphasize the necessity of learning fair representations during pre-training, which is important for equitable model development in medical imaging applications.

The contributions are summarized as follows:

- This study systematically investigates scaling laws in vision medical and natural imaging downstream tasks from a fairness perspective, showing that performance disparities vary according to demographic attributes and dataset.
- We demonstrate that incorporating a fairness loss in the binary cross-entropy loss we achieve a critical point where the loss is constant when scaling model and dataset size.
- We colaborate the idea that scaling alone is insuficient to mitigate bias, highlighting the importance of learning fair representations during pre-training.

## 2 RELATED WORK

**Self-supervised learning.** Scaling laws consistently improve performance across domains as model size, dataset scale, and compute increase Hernandez et al. (2021); Hestness et al. (2017); Kaplan et al. (2020); Zhai et al. (2022); Bahri et al. (2024). Self-supervised learning leverages pretraining tasks to learn relevant representations for diverse downstream tasks without labels, which is essential for scaling datasets and model parameters. The patch-based in the Vision Transformer (ViT) Dosovitskiy et al. (2021) enables Masked Image Modeling (MIM), analogous to BERT Devlin et al. (2019) in text, where masked patches are reconstructed, as in Masked Auto-Encoders (MAE) He et al. (2021). Recent work shows learning representations rather than reconstructing masked patches yields superior outcomes Assran et al. (2023); Garrido et al. (2024). Combining MIM with clustering methods has gained attention due to the complementary aggregation benefits Gidaris et al. (2024); Darcet et al. (2025), while other approaches integrate discriminative losses with MIM objectives to enhance performance Zhou et al. (2022).

**Downstream.** Self-supervised learning models demonstrate exceptional generalization across diverse downstream tasks and domains Azizi et al. (2023); Ali et al. (2024); Siméoni et al. (2025); Venkataramanan et al. (2025). Features learned from pretext tasks facilitate efficient knowledge transfer to target tasks with minimal labeled data, significantly benefiting domains requiring expert annotations Azizi et al. (2023). Most self-supervised learning methods focus on learning local image descriptors, such as pixel reconstruction in MAE He et al. (2021), reflecting the requirements of vision tasks like object detection and segmentation. In contrast, vision-language models emphasize semantic features aligned with text descriptions that capture the overall image content Radford et al. (2021).

**Bias.** Recent investigations into dataset classification problems reveal persistent bias challenges in contemporary machine learning architectures and large datasets Liu & He (2024); Torralba & Efros (2011). This finding proves particularly significant given that bias persistence occurs even when balanced datasets are employed Cui et al. (2024). Furthermore, models exhibit susceptibility to diverse bias sources, including computational complexity, learning sequence, positional encoding, and complexity Lampinen et al. (2024). Complexity features, including semantic content, structural boundaries, and color properties, exhibit strong correlations with object structures and semantic meaning Zeng et al. (2024). Gender bias and other fairness related biases correlate with these identical complexity features Meister et al. (2023).

# 3 METHODS

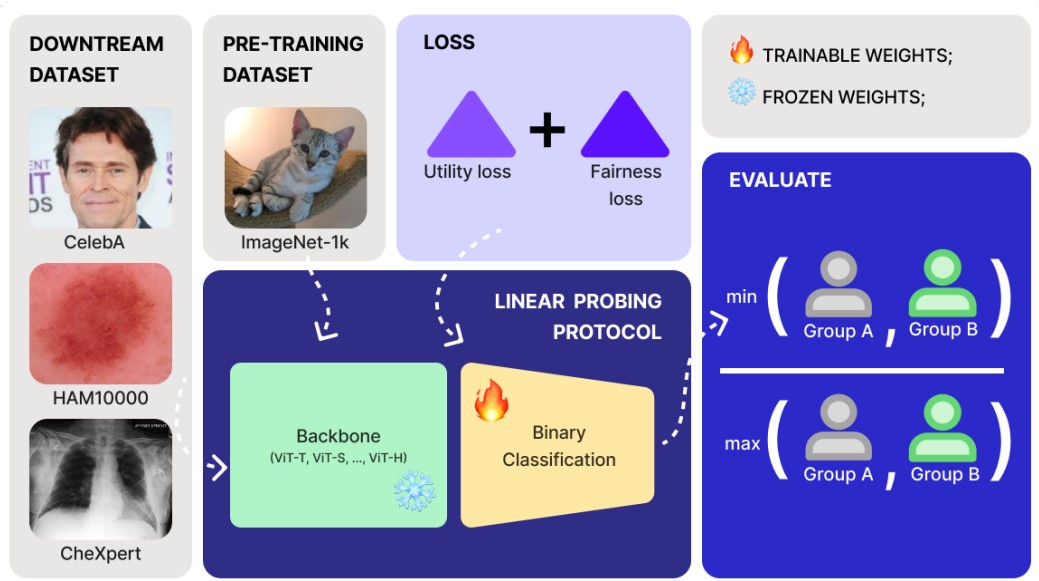

Figure 1: **Overview.** The study systematically evaluated how downscaling dataset size and backbone parameters influence utility and fairness losses. For each task, we sampled a subset of data and trained models using a linear probing protocol. Model performance and fairness metrics were then assessed, with particular attention to disparities across groups defined by metadata attributes such as age and gender. In addition, we examined the effect of incorporating an explicit fairness loss.

This study investigates whether algorithmic fairness can be systematically characterized by scaling laws. Our central research question is to determine if, similar to model performance and binary cross entropy loss, fairness metrics and loss consistently improve as model parameters and dataset sizes increase. To understand how this relationship is affected by domain shift, our evaluation is conducted across distinct image domains: an in-distribution domain, which is closely aligned with the model's pre-training data and two out-of-distribution domain, which is less aligned, allowing us to assess the impact of distributional shifts on fairness outcomes under scaling. An overview of our experimental setup is presented in Figure 1.

**Models.** We employed Hierarchical Vision Transformers (Hiera) Ryali et al. (2023) as our model architecture. The Hiera family encompasses multiple scale variants: tiny (27.1M parameters), small (34.2M parameters), base (50.8M parameters), large (213M parameters) and huge (671M parameters), enabling comprehensive analysis across different model capacities. All models underwent pre-training on ImageNet-1k using MAE as the pretext task without labels or exposure to medical imagery.

We evaluate the model using a linear probing evaluation protocol, whereby the pre-trained model weights remain frozen while training only a linear classifier head using supervised learning with task-specific labels from our target dataset. To investigate inherent bias properties, we deliberately selected models trained exclusively on object-centric datasets, hypothesizing that such training would minimize domain-specific biases; however, our evaluation aimed to determine whether bias manifestations persist even under these controlled pre-training conditions.

**Datasets.** We selected the CelebA Liu et al. (2015) as our natural imaging dataset, wich contains celebrity face images. In the medical imaging we choose two datasets to evaluate disparities manifestations across distinct domains: CheXpert Irvin et al. (2019) and HAM10000 Codella et al. (2019). These datasets were chosen to represent varying degrees of alignment with the pre-trained model's feature representations. CheXpert comprises chest X-ray images presented in grayscale for-

mat, capturing radiological findings across multiple pathological conditions through medical imaging modalities fundamentally different from natural photography, like ImageNet-1k.

HAM10000 contains RGB dermoscopic images of skin lesions, maintaining color information and visual characteristics more closely resembling natural imagery. Although the pre-trained models encountered neither dataset during training, the architectural exposure to RGB images and potential skin representations in ImageNet-1k suggests HAM10000 exhibits greater feature alignment with learned representations compared to CheXpert's grayscale radiological domain. This differential alignment enables investigation of how domain similarity influences bias expression in downstream medical tasks.

**Downstream task.** We implemented binary classification tasks for all datasets to evaluate model performance and disparities manifestations. For CelebA we classified the presence of a smile or not in the images. For CheXpert, we formulated the task as detecting the presence of Edema versus normal findings, creating a clinically relevant diagnostic challenge within chest radiography. In HAM10000, we constructed a malignant versus benign classification framework, where benign cases comprised Melanocytic nevi, Benign keratosis-like lesions, Dermatofibroma, and Vascular lesions, while malignant cases included Melanoma, Basal cell carcinoma, and Actinic keratoses with intraepithelial carcinoma.

**Groups.** We utilize demographic metadata, specifically age and gender, from all datasets to evaluate disparities manifestations across different population groups. For CelebA, we use additional attributes such as mustache, big nose, eyeglasses, and black hair. To assess fairness, we adopt the concept of group fairness, measuring disparities between groups defined by these metadata attributes. For gender classification, we employ the binary categories of male and female, acknowledging the inherent limitations of this approach and its potential exclusion of transgender and non-binary individuals. However, all datasets provide only these two gender categories. Regarding age stratification, we segment the population into three uniform groups spanning the minimum to maximum ages present in each dataset. While this age grouping lacks clinical optimization, it provides sufficient granularity to evaluate performance disparities between demographic classes within our analytical framework.

**Fairness Evaluation.** We employ the Area Under the Receiver Operating Characteristic curve (AUROC) as our primary utility metric to evaluate overall model performance across both tasks, consistent with established benchmarking protocols for these challenges. To quantify fairness disparities between demographic groups, we define $\text{AUROC}_a^{\max}$ as the group $a$ achieving maximum AUROC performance and $\text{AUROC}_b^{\min}$ as the group $b$ with minimum AUROC performance. We calculate the fairness disparity ratio $\alpha$ as:

$$\alpha = \frac{\text{AUROC}_b^{\min}}{\text{AUROC}_a^{\max}}, \tag{1}$$

Where values closer to 1 indicate equitable performance across groups, while lower ratios signify greater disparities manifestation. We denote $\alpha_{AUROC}^{Group}$ as the $\alpha$ value calculated for the group using the AUROC metric.

**Scaling laws.** To investigate scaling laws, we employ cross entropy loss for model training. Consistent with established literature, this loss scales according to the pretraining dataset size, model parameters, for pre-training tasks Kaplan et al. (2020) and downstream tasks Isik et al. (2025), as shown in equation 2, where $A$ and $\beta$ represent coefficients optimized empirically, and $D$ denotes the downstream dataset size. This formulation extends to model parameters by substituting $D$ with $P$, where $P$ is the number of parameters in the model, thereby maintaining identical scaling dynamics

$$L(D) = \frac{A}{D^\beta} + E \tag{2}$$

**Hyperparameters.** For all experiments, a consistent set of hyperparameters was maintained to ensure comparability across trials. The batch size was set to 256, and training proceeded for 30

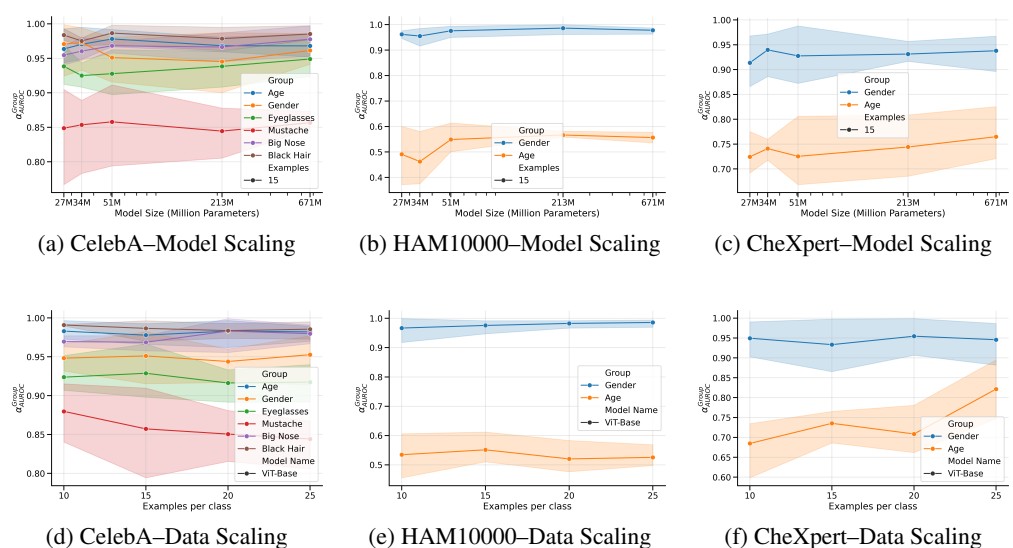

(a) CelebA–Model Scaling  (b) HAM10000–Model Scaling  (c) CheXpert–Model Scaling

(d) CelebA–Data Scaling  (e) HAM10000–Data Scaling  (f) CheXpert–Data Scaling

Figure 2: **Fairness metrics across datasets.** We evaluate the $\alpha_{AUROC}^{Group}$ for the groups in the datasets, where the metric exhibits different scaling behaviors depending on the dataset and attribute. Results are averaged over three independent seeds, and the error bars represent the 95% confidence interval.

epochs, a duration found sufficient for model convergence. We employed the AdamW optimizer with a learning rate of 0.0008, incorporating a cosine warmup schedule. To fit the scaling law coefficients, we utilized the same methodology as prior work Isik et al. (2025); Hoffmann et al. (2022), employing the Huber loss Huber (1964) and optimizing with the L-BFGS algorithm Nocedal (1980), further methodological details are provided in the appendix B and C.

## 4 RESULTS AND DISCUSSION

### 4.1 UTILITY LOOK

The first set of analyses evaluated fairness metrics using a binary cross-entropy utility loss in the downstream binary classification tasks on the CelebA, HAM10000, and CheXpert datasets. As shown in Figure 2, the datasets exhibited divergent patterns. Specifically, the CheXpert dataset demonstrated an improvement in $\alpha_{AUROC}^{age}$ when both the dataset and model were scaled, whereas for HAM10000 the $\alpha_{AUROC}^{age}$ remained largely constant across evaluations. The CelebA dataset includes attributes, such as mustache, that reduce fairness metrics, while other attributes show no clear improvement. These results indicate that the impact of scaling on fairness is dependent on the dataset.

**Protected attributes.** An important observation is that the scaling behavior differed across protected attributes. The $\alpha_{AUROC}^{gender}$ achieved consistently high values, approaching 1, in all datasets. By contrast, the $\alpha_{AUROC}^{age}$ yielded a mean value of 0.96 for CelebA, 0.7 for CheXpert, and 0.5 for HAM10000. These results highlight that fairness performance varies substantially across protected attributes.

This finding is consistent with previous research showing that in medical image embeddings, gender attributes tend to have a more global representation, whereas age is represented in a more localized manner Queiroz et al. (2025a); Graf et al. (2024), a similar pattern is observed in medical benchmark for others datasets in supervised and self-supervised models Zong et al. (2023); Jin et al. (2024). These results suggest that gender captures broader structural signals across the datasets, while age distinctions may emerge only within specific subgroups, such as males aged 20 and females aged 20. In CelebA, the age attribute was divided into young and non-young, providing a more general representation than in CheXpert and HAM10000, where it was split into three groups.

Table 1: **Utility results.** We evelute the datasets focusing on accuracy (ACC), AUROC, and binary cross-entropy loss are presented for CelebA, HAM10000, and CheXpert datasets. The reported values correspond to the mean performance over three independent seeds.

| Config | CelebA | | | HAM10000 | | | CheXpert | | |
|---|---|---|---|---|---|---|---|---|---|
| | AUROC ↑ | ACC ↑ | Loss ↓ | AUROC ↑ | ACC ↑ | Loss ↓ | AUROC ↑ | ACC ↑ | Loss ↓ |
| *Data Scale - ViT-Base* | | | | | | | | | |
| 10 | 0.7109 | 0.6219 | 0.6550 | 0.843 | 0.810 | 0.542 | 0.611 | 0.660 | 0.629 |
| 15 | 0.7124 | 0.6271 | 0.6512 | 0.872 | 0.821 | 0.524 | 0.642 | 0.721 | 0.604 |
| 20 | 0.7637 | 0.6547 | 0.6413 | 0.873 | 0.834 | **0.509** | 0.667 | 0.711 | 0.613 |
| 25 | **0.7681** | **0.6602** | **0.6373** | **0.880** | **0.840** | 0.511 | **0.671** | **0.745** | **0.599** |
| *Model Scale - 15 examples per class* | | | | | | | | | |
| ViT-Tiny | 0.6655 | 0.6090 | 0.6682 | 0.875 | 0.823 | 0.561 | **0.665** | **0.752** | 0.619 |
| ViT-Small | 0.7101 | **0.6301** | 0.6585 | 0.846 | 0.800 | 0.569 | 0.610 | 0.728 | 0.626 |
| ViT-Base | **0.7120** | 0.6234 | **0.6524** | 0.870 | 0.821 | 0.527 | 0.647 | 0.725 | 0.600 |
| ViT-Large | 0.6994 | 0.6223 | 0.6525 | 0.900 | **0.869** | 0.464 | 0.651 | 0.729 | **0.581** |
| ViT-Huge | 0.6856 | 0.6262 | 0.6555 | **0.910** | 0.867 | **0.421** | 0.646 | 0.699 | 0.591 |

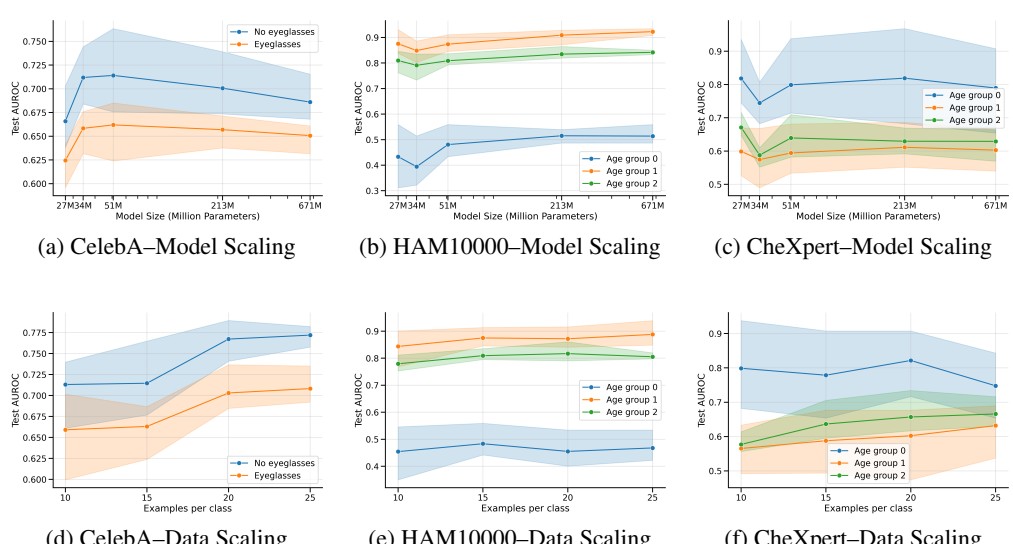

(a) CelebA–Model Scaling   (b) HAM10000–Model Scaling   (c) CheXpert–Model Scaling

(d) CelebA–Data Scaling   (e) HAM10000–Data Scaling   (f) CheXpert–Data Scaling

Figure 3: **Group AUROC metrics.** We evaluate the AUROC for each selected group in the datasets. When scaling model parameters, we use only 15 examples per class, whereas when scaling the dataset size, the ViT-Base model is employed. Results are averaged over three independent seeds, with error bars representing the 95% confidence interval.

**Unfairness and Local Representations.** Our findings suggest that unfairness primarily arises from local representations. Subgroup distinctions are particularly relevant for unfairness outcomes Alloula et al. (2025); Bissoto et al. (2025); Queiroz et al. (2025a). Scaling both the dataset and model parameters was more effective in mitigating unfairness linked to local representations in less alignment datasets, as global representations already exhibited fair outcomes (Figure 2). Similarly, in adversarial learning, models suppress unwanted features associated with protected variables, thereby promoting a more global representation Wang et al. (2019).

**Differences Between Datasets.** Consistent with prior findings, greater distributional alignment within a dataset corresponds to improved scaling of utility metrics and loss Isik et al. (2025). As shown in Table 1, both accuracy and AUROC increased across datasets. However, for the dataset with less alignment, CheXpert, the ViT-Tiny model achieved the highest accuracy and AUROC. These results underscore the influence of dataset characteristics on model performance.

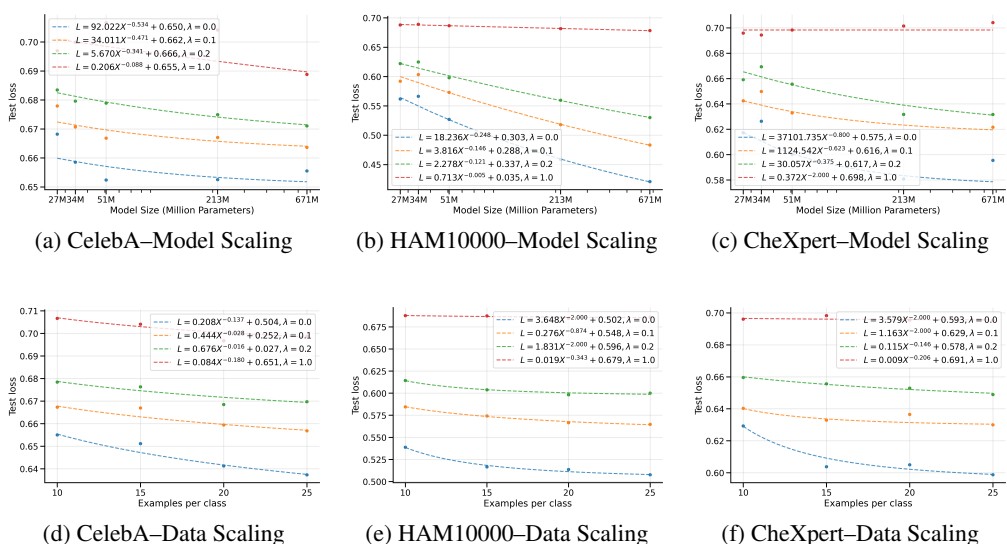

Figure 4: **Scaling Laws for utility loss and fairness loss.** These plots illustrate how the cross-entropy loss, combined with a fairness loss, varies as additional fairness constraints are incorporated into the overall loss, highlighting the critical point at which the loss stabilizes. Data points represent the mean values over three independent seeds.

**Subgroup Performance.** Fairness metrics exhibited divergent patterns across datasets. For the less aligned CheXpert dataset, scaling both the data and model parameters led to improved fairness metrics. Conversely, for CelebA and HAM10000, scaling the data had little effect, while scaling the model parameters yielded modest improvements (Figure 2). A detailed analysis of age groups (Figure 3) reveals that the improvement in CheXpert was accompanied by a decrease in the group with the highest AUROC. In contrast, CelebA and HAM10000 showed uniform improvements across all groups. These findings highlight the complex interaction between dataset alignment, scaling, and subgroup fairness.

Notably, the most underrepresented group, age group 0 in HAM10000, shows the lowest AUROC performance. In contrast, the same age group 0 in CheXpert, also the most underrepresented, achieves the highest AUROC. For detailed dataset distributions, see Appendix A.

**Algorithmic bias.** A possible explanation is that the features achieves better dataset alignment representation, as reflected in the higher metrics shown in Table 1. However, improved overall metrics often coincide with increased unfairness Zhao & Gordon (2022); Wei & Niethammer (2021), particularly for underrepresented groups; for example, while the most represented group attains an AUROC of 0.85, the underrepresented group reaches only 0.5 (Figure 3). Previous studies have demonstrated that supervised models tend to learn features biased toward demographic attributes to optimize utility metrics Stanley et al. (2025). Moreover, self-supervised models appear to capture associations with other attributes despite not explicitly relying on labels or attributes Wang et al. (2024). These findings highlight the complex trade-off between utility and fairness in model training.

An implication of this finding is that in datasets with less aligned distributions, models may not learn specific biases related to frequency, color, shape, or other dataset characteristics Lampinen et al. (2024); Zeng et al. (2024); Meister et al. (2023). Consequently, scaling the dataset and model parameters has a more pronounced impact on fairness in these scenarios. Models that learn more generalizable features demonstrate improved fairness and scale more effectively with increasing parameters and dataset size (Figure 2). However, in terms of utility, adopting a generalist approach does not necessarily lead to improved performance metrics (Table 1).

## 4.2 WHEN WE COMBINED A FAIRNESS LOSS

The previous experiment employed a cross-entropy loss, a utility-focused objective function optimized without accounting for protected attributes. In this analysis, we introduce a fairness loss component normalized by age and eyeglasses to improve the $\alpha_{AUROC}^{Age}$ for CheXpert, HAM10000, and $\alpha_{AUROC}^{Eyeglasses}$ for CelebA. The combined loss function is defined as follows:

$$\mathcal{L}_{\text{total}} = \mathcal{L}_{ce} + \lambda \cdot \mathcal{L}_{\text{fair}}, \tag{3}$$

where $\mathcal{L}_{ce}$ denotes the cross-entropy loss, $\mathcal{L}_{\text{fair}}$ represents the fairness loss penalizing the p-norm of the violation vector Buyl et al. (2024), and $\lambda \in [0, 1]$ regulates the relative importance of the fairness term.

**Scaling Laws.** The effects of incorporating the fairness loss are presented in Figure 4. The binary cross-entropy loss exhibits the same behavior described in Equation 2, consistent with previous studies Isik et al. (2025); Hoffmann et al. (2022). The key finding is that as the weight of the fairness component $\lambda$ increases, the total loss reaches a critical point after which it plateaus, indicating a convergence to a stable fairness-utility trade-off in the loss even with the scale of the dataset and model parameters.

**We need a fair loss in the pre-training phase.** Consistent with previous findings, unfairness primarily stems from local representations such as age. In this context, we target these representations by penalizing the p-norm of the violation vector in the fairness loss Buyl et al. (2024). The results demonstrate that these local subgroups pose a significant challenge in both datasets. These observations underscore the critical importance of incorporating a fairness loss during pre-training to address disparities effectively.

Focusing solely on local image descriptors techniques, such as MAE He et al. (2021) and pixel matching in videos Jabri et al. (2020) using a utility loss, does not enhance fairness when scaling. While incorporating fairness into pretraining remains challenging due to the scarcity of demographic attribute at scale. Recent studies show that data curation through feature-based clustering and balanced sampling improves model performanceVo et al. (2024); Queiroz et al. (2025a); Siméoni et al. (2025); these scalable methods do not explicitly address sensitive attributes. Moreover, predicting clustered local patches in pretraining tasks, as demonstrated by CAPI Darcet et al. (2025), enhances dense representation quality. Combining this with clustering at the image level may improve both fairness and utility outcomes.

## 4.3 THE TRADE-OFF BETWEEN FAIRNESS AND UTILITY

As previously discussed, models exhibit a trade-off between fairness and utility. The preceding section examined how increasing the fairness loss weight influences the overall loss. In this section, we investigate the effect of the loss function defined in Equation 3 on $\alpha_{AUROC}^{Eyeglasses}$ for CelebA and $\alpha_{AUROC}^{Age}$ for CheXpert and HAM10000.

**Pareto Front.** The pareto front highlights distinct scaling behaviors across domains (Figure 5). In alignment datasets, scaling dataset size produces consistent improvements in both fairness and utility up to approximately 15 examples per class. Conversely, scaling model parameters reveals domain-specific trends: in CelebA, a pronounced trade-off emerges between utility and fairness, where improvements in one dimension reduce performance in the other; whereas in HAM10000, enlarging model capacity yields concurrent gains in both fairness and utility up to the ViT-L.

In comparison with the less aligned dataset, scaling the dataset size benefits both fairness and utility (Figure 2). The results suggest that, for less aligned datasets, increasing dataset size has the potential to enhance fairness while also improving utility. However, when scaling model parameters, a trade-off emerges, indicating that gains in one dimension may come at the expense of the other.

Considering age and eyeglasses as groups that are more localized in the representation space, the evaluation of $\alpha_{AUROC}^{Age}$ and $\alpha_{AUROC}^{Eyeglasses}$ reflects the extent to which disparities between groups are

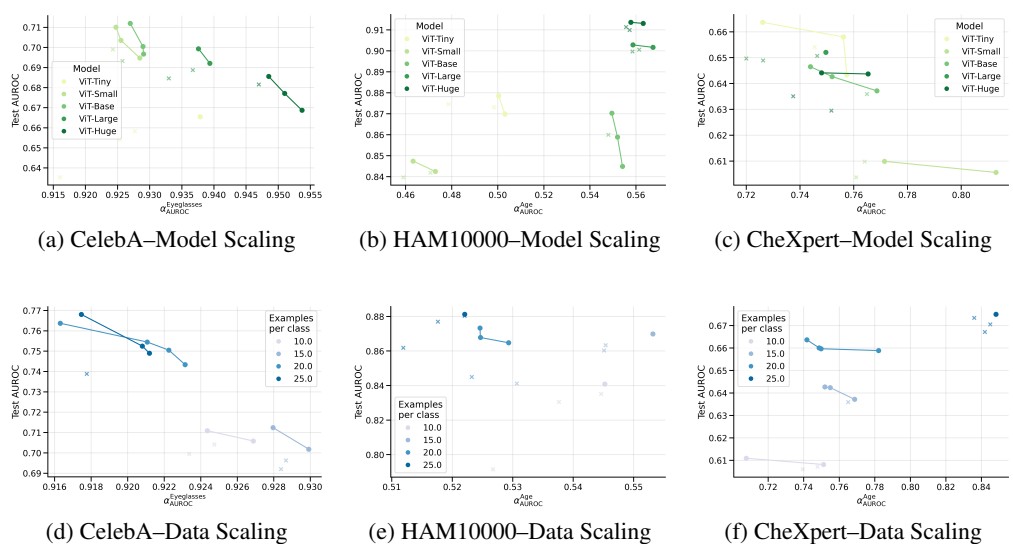

Figure 5: **Pareto Fronts for Fairness-Utility Trade-off.** The Pareto front illustrates the optimal balance between fairness and utility when scaling model size and data size, respectively. The points represents the mean values over three independent seeds. Each point in the plot correspond a different value of $\lambda \in [0, 0.1, 0.2, 1]$.

mitigated. When examining the overall representation, we use AUROC as a global measure of performance. Our results show that in more aligned datasets, improvements in AUROC often coincide with increased disparities between groups. However, there are cases where both reduced disparities and higher AUROC can be achieved, highlighting the distinction between global and local representations.

**Improve features for all subgroups.** The results underscore the critical importance of learning more effective local and global representations. Local representations are more difficult to improve, and from this perspective, we encourage methods that prioritize enhancing local representations rather than merely equalizing local and global ones. Improving local representations to mitigate disparities could be a more beneficial way to advance fairness Sabuncu et al. (2025). This aligns with the notion that strategies aimed at strengthening the entire representation space are essential. Consistent with this perspective, recent research on knowledge agglomeration, exemplified by RA-DIO Heinrich et al. (2025); Ranzinger et al. (2024), which integrates features from diverse models such as CLIP Radford et al. (2021), DINOv2 Oquab et al. (2023), and SAM Kirillov et al. (2023), offers promising avenues for enhancing representation quality across the entire space.

## 5 CONCLUSION

This study systematically investigated the prevailing assumption that scaling laws, which consistently improve model performance, would similarly mitigate fairness disparities across protected groups. Our findings reveal a complex and dataset-dependent relationship, challenging the notion that simply increasing model and data size is not a sufficient strategy for achieving fairness. Notably, while scaling enhances fairness for certain protected groups, particularly when employing a fairness-targeted loss, it reaches a critical threshold beyond which gains plateau, underscoring the insufficiency of scale alone in mitigating bias. These findings emphasize the imperative of integrating fairness considerations during pre-training to address local representation disparities effectively. Future work should explore advanced methods for learning equitable representations in diverse domains to realize truly fair and generalizable AI models.

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

APPENDIX

## A  DATASET DETAILS

This appendix provides descriptive statistics of the datasets employed in our experiments. We report demographic distributions (age, gender), diagnostic labels, and age-related summary measures for both the full datasets and subsampled fractions.

Table 2: Summary statistics of CheXpert dataset across sampled fractions

| Fraction | Age (%) | | | Gender (%) | | Edema (%) | | Age (years) | | | |
|---|---|---|---|---|---|---|---|---|---|---|---|
| | Group 0 | Group 1 | Group 2 | Male | Female | Negative | Positive | Mean | Std | Min | Max |
| all dataset | 9.49 | 51.12 | 39.39 | 59.37 | 40.63 | 70.80 | 29.20 | 60.43 | 17.82 | 18 | 90 |
| 10 | 5.00 | 50.00 | 45.00 | 35.00 | 65.00 | 50.00 | 50.00 | 63.80 | 15.34 | 29 | 86 |
| 15 | 6.67 | 50.00 | 43.33 | 43.33 | 56.67 | 50.00 | 50.00 | 63.10 | 16.23 | 18 | 86 |
| 20 | 5.00 | 52.50 | 42.50 | 47.50 | 52.50 | 50.00 | 50.00 | 63.25 | 15.88 | 18 | 87 |
| 25 | 4.00 | 56.00 | 40.00 | 44.00 | 56.00 | 50.00 | 50.00 | 62.24 | 15.92 | 18 | 87 |

Table 3: Summary statistics of HAM10000 dataset across sampled fractions

| Fraction | Age (%) | | | Gender (%) | | Malignant (%) | | Age (years) | | | |
|---|---|---|---|---|---|---|---|---|---|---|---|
| | Group 0 | Group 1 | Group 2 | Male | Female | Benign | Malignant | Mean | Std | Min | Max |
| all dataset | 12.07 | 66.05 | 21.87 | 54.51 | 45.49 | 78.51 | 21.49 | 52.01 | 17.41 | 1 | 85 |
| 10 | 15.00 | 55.00 | 30.00 | 55.00 | 45.00 | 50.00 | 50.00 | 53.00 | 19.49 | 25 | 85 |
| 15 | 10.00 | 53.33 | 36.67 | 50.00 | 50.00 | 50.00 | 50.00 | 56.50 | 18.06 | 25 | 85 |
| 20 | 10.00 | 52.50 | 37.50 | 47.50 | 52.50 | 50.00 | 50.00 | 55.75 | 19.53 | 5 | 85 |
| 25 | 8.00 | 56.00 | 36.00 | 54.00 | 46.00 | 50.00 | 50.00 | 57.00 | 18.46 | 5 | 85 |

Table 4: Summary statistics of CelebA dataset across sampled fractions

| Fraction | Age (%) | | Gender (%) | | Eyeglasses (%) | | Mustache (%) | | Big Nose (%) | | Black Hair (%) | | Task (Smiling) (%) | |
|---|---|---|---|---|---|---|---|---|---|---|---|---|---|---|
| | Non young | Young | Female | Male | 0 | 1 | 0 | 1 | 0 | 1 | 0 | 1 | 0 | 1 |
| all dataset | 22.11 | 77.89 | 58.06 | 41.94 | 93.54 | 6.46 | 95.92 | 4.08 | 76.44 | 23.56 | 76.10 | 23.90 | 52.03 | 47.97 |
| 10 | 25.00 | 75.00 | 65.00 | 35.00 | 90.00 | 10.00 | 100.00 | 0.00 | 65.00 | 35.00 | 85.00 | 15.00 | 50.00 | 50.00 |
| 15 | 23.33 | 76.67 | 60.00 | 40.00 | 93.33 | 6.67 | 100.00 | 0.00 | 70.00 | 30.00 | 86.67 | 13.33 | 50.00 | 50.00 |
| 20 | 22.50 | 77.50 | 60.00 | 40.00 | 92.50 | 7.50 | 97.50 | 2.50 | 75.00 | 25.00 | 85.00 | 15.00 | 50.00 | 50.00 |
| 25 | 22.00 | 78.00 | 58.00 | 42.00 | 94.00 | 6.00 | 98.00 | 2.00 | 80.00 | 20.00 | 84.00 | 16.00 | 50.00 | 50.00 |

In addition to the distributional statistics, we report the overall size and partitioning protocol for each dataset.

The CheXpert dataset contains 224,316 chest radiographs from 65,240 patients. Following prior work, we utilized the official split provided by the dataset authors: 223,410 images for training (we sample the examples here), 234 validation, and 234 images for test that are in the official dataset. Even thought the test set is small, it is the only one with the high confidenty label on the annotations.

The HAM10000 dataset comprises 10,015 dermatoscopic images from different populations and acquisition modalities. We randomly divided the dataset into 8,307 training, 557 validation, and 1084 testing.

The CelebA dataset consists of 202,599 celebrity face images annotated with 40 binary attributes. We adopted the standard partition provided by the dataset creators: 162,770 training images, 994 validation images, and 18,873 test images.

## B  IMPLEMENTATION DETAILS

**Linear probing.** The parameters used in the linear probing experiments are reported in Table 5. Identical settings were applied across all datasets and models. To account for variability, we conducted experiments with three random seeds, each defining a different training subset. Details of the computational setup and code implementation will be provided after the review.

| config | value |
|---|---|
| optimizer | AdamW |
| learning rate | 8e-4 |
| weight decay | 0 |
| optimizer momentum | $\beta_1, \beta_2 = 0.9, 0.999$ |
| learning rate scheduler | cosine decay |
| warmup epochs | 5 |
| loss | cross entropy |
| label smoothing | 0.1 |
| accumulate grad batches | 1 |
| training epochs | 30 |
| batch size | 256 |
| augment train | RandomResizedCrop |
| freeze backbone | true |
| seed | 42, 44, 52 |

Table 5: Linear probing setting.

## C  SCALING LAWS.

The scaling laws were calculated using a Huber loss function for robustness against outliers. The optimization process involved multiple random initializations to find the best fit for the scaling law parameters. The specific configuration used for fitting the scaling laws is detailed in Table 6.

| config | value |
|---|---|
| Loss function | Huber Loss |
| Huber loss delta ($\delta$) | 1e-3 |
| Optimization method | L-BFGS-B |
| Number of initializations | 50 |
| Training data ratio | 1.0 |
| Parameter bounds (A, $\beta$, E) | A, E $\in (10^{-6}, \infty)$, $\beta \in (10^{-6}, 2.0)$ |
| Optimizer max iterations | 1000 |
| Optimizer function tolerance (ftol) | 1e-9 |
| Optimizer gradient tolerance (gtol) | 1e-5 |

Table 6: Configuration settings for fitting the scaling laws.

| Dataset | $\lambda$ | A | $\beta$ | E | Huber loss |
|---|---|---|---|---|---|
| CheXpert | 0.0 | 3.5792 | 2.0000 | 0.5933 | 7.226e-06 |
| CheXpert | 0.1 | 1.1632 | 2.0000 | 0.6285 | 4.977e-06 |
| CheXpert | 0.2 | 0.0844 | 0.4509 | 0.6302 | 1.188e-06 |
| CheXpert | 1.0 | 0.0081 | 0.5917 | 0.6944 | 1.898e-06 |
| HAM10000 | 0.0 | 3.6481 | 2.0000 | 0.5021 | 3.156e-06 |
| HAM10000 | 0.1 | 0.2761 | 0.8742 | 0.5477 | 1.102e-06 |
| HAM10000 | 0.2 | 1.8309 | 2.0000 | 0.5958 | 2.488e-06 |
| HAM10000 | 1.0 | 0.0192 | 0.3431 | 0.6792 | 1.067e-06 |
| CelebA | 0.0 | 0.2078 | 0.1369 | 0.5038 | 3.625e-06 |
| CelebA | 0.1 | 0.4438 | 0.0285 | 0.2522 | 3.610e-06 |
| CelebA | 0.2 | 0.6761 | 0.0157 | 0.0266 | 4.018e-06 |
| CelebA | 1.0 | 0.0845 | 0.1802 | 0.6512 | 3.770e-06 |

Table 7: Values of the parameters obtained by fitting the scaling law equation for dataset size scaling.

## D  UTILITY LOOK

Here we report more examples of the utility look as showed in the Figure 3 in the Figure 6 for HAM10000 and CheXpert datasets and in the Figure 7 for CelebA dataset.

| Dataset | $\lambda$ | A | $\beta$ | E | Huber loss |
|---|---|---|---|---|---|
| CheXpert | 0.0 | 37101.73 | 0.79996 | 0.57540 | 3.43e-05 |
| CheXpert | 0.1 | 1124.54 | 0.62273 | 0.61603 | 2.23e-05 |
| CheXpert | 0.2 | 30.05744 | 0.37527 | 0.61657 | 2.04e-05 |
| CheXpert | 1.0 | 0.37189 | 2.00000 | 0.69830 | 1.34e-05 |
| HAM10000 | 0.0 | 18.23617 | 0.24796 | 0.30258 | 1.88e-05 |
| HAM10000 | 0.1 | 3.81551 | 0.14630 | 0.28802 | 2.11e-05 |
| HAM10000 | 0.2 | 2.27826 | 0.12145 | 0.33730 | 1.29e-05 |
| HAM10000 | 1.0 | 0.71316 | 0.00504 | 0.03450 | 9.37e-07 |
| CelebA | 0.0 | 92.02169 | 0.53387 | 0.65002 | 1.56e-05 |
| CelebA | 0.1 | 34.01101 | 0.47099 | 0.66169 | 8.42e-06 |
| CelebA | 0.2 | 5.66999 | 0.34080 | 0.66590 | 2.08e-06 |
| CelebA | 1.0 | 0.20644 | 0.08815 | 0.65530 | 2.19e-05 |

Table 8: Values of the parameters obtained by fitting the scaling law equation for model parameter scaling.

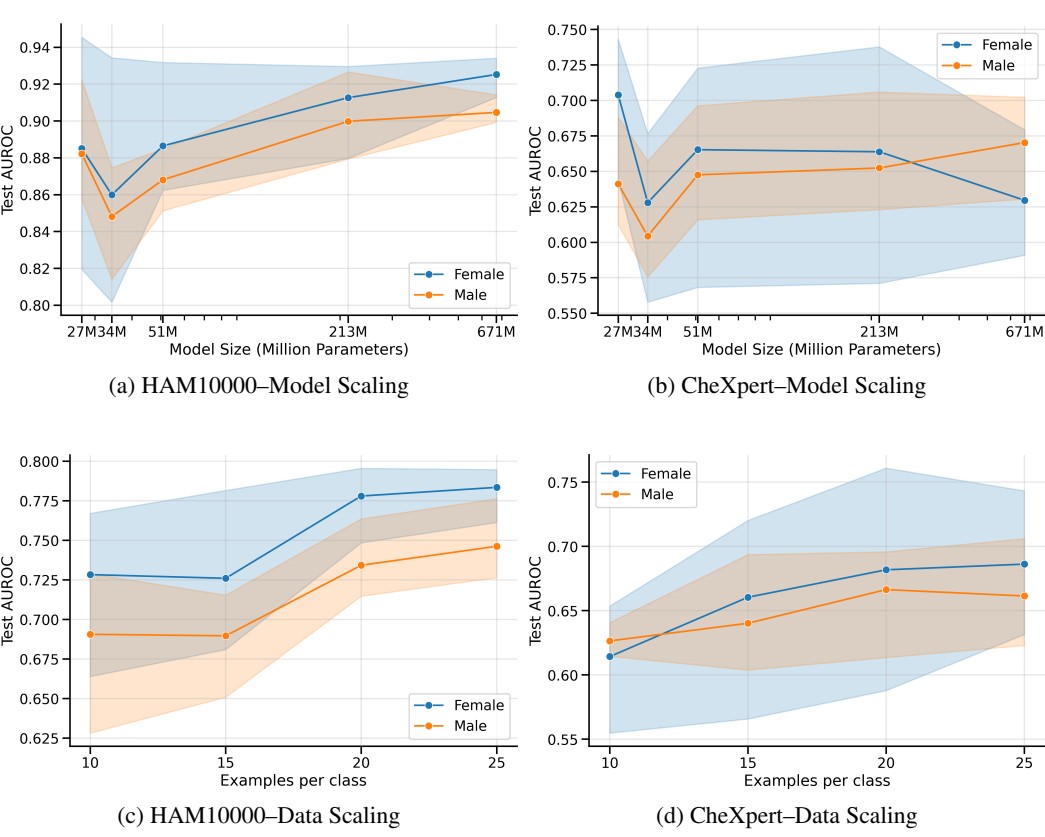

(a) HAM10000–Model Scaling

(b) CheXpert–Model Scaling

(c) HAM10000–Data Scaling

(d) CheXpert–Data Scaling

Figure 6: **Group AUROC metrics.** Evaluation of the AUROC for the gender groups in the CheXpert and HAM10000. When scaling model parameters, we use only 15 examples per class, whereas when scaling the dataset size, the ViT-Base model is employed. Results are averaged over three independent seeds, with error bars representing the 95% confidence interval.

# E  LARGE LANGUAGE MODELS (LLMS) IN PAPER WRITING

We utilize LLMs to correct grammar and improve the text fluency.

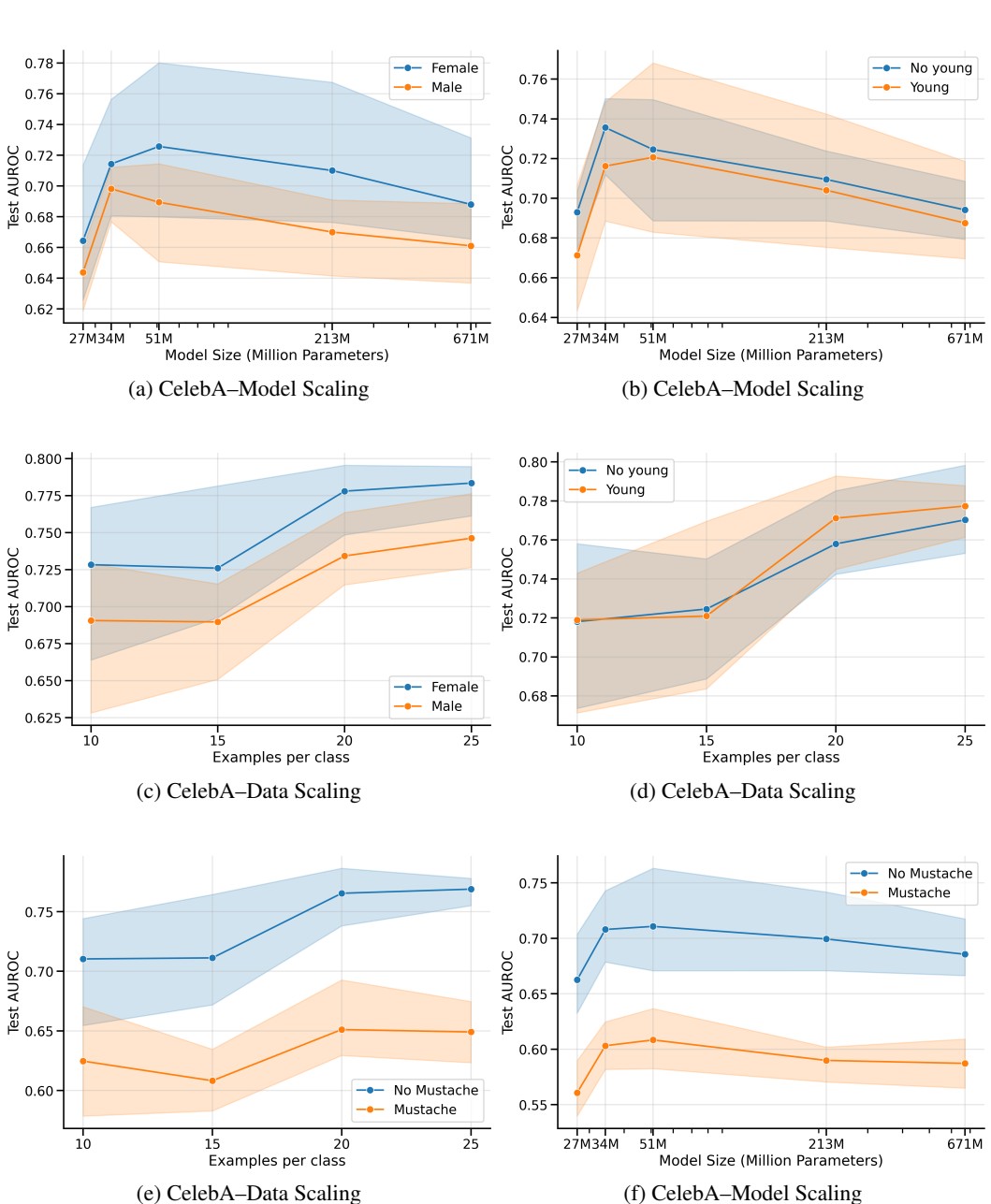

Figure 7: **Gender AUROC metrics.** Evaluation of the AUROC for the gender, age and mustache groups in the CelebA dataset. When scaling model parameters, we use only 15 examples per class, whereas when scaling the dataset size, the ViT-Base model is employed. Results are averaged over three independent seeds, with error bars representing the 95% confidence interval.

