# OpenReview forum: "The Limits of Fairness Gains Under Scaling in Vision Models"
_ICLR.cc/2026/Conference — Submitted to ICLR 2026_

### Official Review · Reviewer_iS83 · 2025-10-18

**Soundness:** 3
**Presentation:** 3
**Contribution:** 2
**Rating:** 4
**Confidence:** 4

**Summary:**

This paper investigates the relationship between scaling (of model parameters and dataset size) and fairness in vision models. The authors empirically test the hypothesis that the "scaling laws" which lead to improved model performance (utility) will also lead to improved fairness. Using three distinct datasets (CelebA, HAM10000, and CheXpert) and a family of Hierarchical Vision Transformers (Hiera), they find that scaling alone is not a sufficient strategy for mitigating bias.

**Strengths:**

1. The topic is timing, as more data and larger models are adopted currently.

2. The experiments are extensive.

3. The overall presentation is clear and easy to follow.

**Weaknesses:**

1. The paper introduces a joint loss function $\mathcal{L}_{total}=\mathcal{L}_{ce}+\lambda\cdot\mathcal{L}_{fair}$. But the specific $\mathcal{L}_{fair}$ used (penalizing the p-norm of the violation vector) is just one of many possible fairness interventions.

2. A primary limitation of the current study is its focus on scaling the downstream dataset size instead of the pre-training dataset scale.  The paper would be strengthened by considering pre-trained models and datasets, rather than just sub-sampling the downstream task data.

3. The paper's reliance on a single, primary fairness metric limits the scope of its conclusions. Please include more widely adopted fairness notions.

4. The paper correctly notes that model scaling produces divergent fairness trends across datasets (e.g., improving 'Age' fairness on CheXpert while having no effect on HAM10000 , and even worsening 'Mustache' fairness on CelebA ). The primary explanation offered hinges on the concept of "dataset alignment" and a distinction between "local" and "global" representations. This explanation feels incomplete and not convincing enough.

5. Regarding data scaling, the results depend heavily on how the data is sampled. Different sampling methods would create different data distributions, which could easily change the outcome. This makes it hard to know if the reported effects are from scaling or just different data distributions.

**Questions:**

1. The models were pre-trained on ImageNet-1k. How do the known biases within ImageNet-1k (e.g., geographic, demographic) interact with the downstream tasks? Would they influence the experiment results?

2. The paper suggests unfairness arises from "local representations" while "global representations" are fairer. This is an interesting but feels somewhat speculative and could be defined more concretely. What properties of the model or data make one attribute "local" and another "global"?

---

> ### Author Response · Authors · 2025-11-21
>
> We thank the reviewer for highlighting the timeliness of our topic, the scope of our experiments, and the clarity of our presentation.
>
> > W1.
>
> This loss targets our main argument by penalizing group-level fairness violations. We will also add other fairness mitigation techniques to broaden the discussion.
>
> > W2.
>
> We acknowledge the limitation of scaling only the downstream dataset size, not the pretraining dataset size. Finding models that vary both parameters and pretraining data is challenging and computationally costly. Our focus is on downstream tasks where fairness is critical and interventions are practical. Recent research supports downstream scaling laws [1]. Extending scaling to pretraining datasets is an important future work.
>
> > W3.
>
> We extended our evaluation to include popular fairness metrics like Equal Opportunity and Demographic Parity.
>
> > W4.
>
> The results shown in the paper could support the explanation regarding local and global representations, given that curating data based on clusters equally formed in the model’s representation space improves performance [2]. Recent DINOv3 models employ this method among others to achieve better results [3]. We believe the performance gains in [2, 3] may be related to improved local representation. The paper [4] shows that the clusters formed by the method in [2] represent metadata of medical images, and that these sensitive attributes have different “local” and “global” representations [5].
>
> We agree that this explanation is incomplete. Understanding what makes an attribute local vs. global will strengthen this. Future work will explore this to improve fairness in self-supervised learning, as suggested by [2].
>
> > W5.
>
> To address potential variability, we conducted experiments using three different random seeds to sample subsets, thereby capturing different possible data distributions.
>
> > Q1.
>
> We selected the datasets and pretrained models to investigate how known biases in ImageNet-1k interact with downstream tasks. We hypothesize that learned representations inherit biases related to complexity, data imbalance, color, object types, and other factors[6] and these biases have implications in fairness and utility metrics. For example, CelebA is closely aligned with ImageNet in terms of color and object-centric features, and accordingly, we observe stronger downstream unfairness as shown in the updated results in Table 1 of the revised manuscript.
>
> Conversely, for less aligned datasets such as HAM10000 (medical domain), which consists of RGB images potentially biased by color frequency, our models demonstrate good utility and fairness metrics. In more distinct domains like CheXpert, with different imaging modalities and grayscale, we achieve fair outcomes, but utility metrics are lower. This suggests that domain alignment influences bias transfer and downstream utility and fairness metrics.
>
> > Q2.
>
> The question of what properties make one attribute “local” and another “global” is a good one. Recent works show that data curation methods that better sample representative data in the domain improve model performance [2, 3]. Their algorithms select data by balancing global and local groups equally in the representation space [2]. In our work, we hypothesize that the limitation in scaling fairness (Figure 4) arises because pretraining optimizes global representations more easily, while local representations face more difficulties due to bias, noise, and complexity [6].
>
> In Figure 4, we show that CelebA, which we suppose is the most aligned dataset and suffers more from bias in ImageNet data, is already able to optimize fairness better than less aligned datasets, even though its fairness metrics are worse (Table 1). We suggest that we have a complex challenge in bias learned from data and representation learning, as explored in [7]. We suggest that understanding the representation space and optimizing across both global and local aspects is a promising direction for future work in data curation and representation learning.
>
> References:
>
> [1] B. Isik, N. Ponomareva, H. Hazimeh, D. Paparas, S. Vassilvitskii, and S. Koyejo, “Scaling Laws for Downstream Task Performance in Machine Translation,”.
>
> [2] H. V. Vo *et al.*, “Automatic Data Curation for Self-Supervised Learning: A Clustering-Based Approach,”.
>
> [3] O. Siméoni *et al.*, “DINOv3,”.
>
> [4] D. Queiroz, A. Anjos, and L. Berton, “Using Backbone Foundation Model for Evaluating Fairness in Chest Radiography Without Demographic Data,”.
>
> [5] R. Graf *et al.*, “Detecting Unforeseen Data Properties with Diffusion Autoencoder Embeddings using Spine MRI data,”.
>
> [6] A. K. Lampinen, S. C. Y. Chan, and K. Hermann, “Learned feature representations are biased by complexity, learning order, position, and more,”.
>
> [7] Z. Liu and K. He, “A Decade’s Battle on Dataset Bias: Are We There Yet?,”

---

> > ### Comment · Reviewer_iS83 · 2025-11-22
> >
> > Thank you for the detailed reply. While I appreciate the effort to address my concerns, the proposed revisions are extensive. As a result, I will stand by my original rating.

---

### Official Review · Reviewer_5tEA · 2025-10-31

**Soundness:** 2
**Presentation:** 3
**Contribution:** 2
**Rating:** 4
**Confidence:** 3

**Summary:**

This work investigates how scaling, increasing model size and dataset volume, affects fairness in vision models across medical and natural imaging domains. Using Hierarchical Vision Transformers (Hiera) pretrained on ImageNet-1k and evaluated via linear probing on CelebA, CheXpert, and HAM10000, the authors measure fairness through AUROC disparity ratios across demographic groups (e.g., age, gender). They find that scaling improves fairness only up to a point, with diminishing returns beyond a critical threshold, especially when a fairness-aware loss (combining cross-entropy and a fairness penalty) is used. Fairness gains are highly dependent on the protected attribute and dataset domain: gender disparities are consistently low, while age-related disparities vary significantly and are harder to mitigate due to their localized representational nature. Crucially, the study demonstrates that scaling alone cannot eliminate bias, underscoring the need to integrate fairness considerations, particularly during pre-training, to address structural inequities.

**Strengths:**

1. The paper presents a systematic and empirical investigation into the relationship between scaling, both in model size and dataset volume, and fairness in vision models, addressing a critical gap in the literature.

2. This work provides nuanced analyses of fairness across different protected attributes and data domains.

**Weaknesses:**

1. One notable weakness of the study is its reliance on linear probing with frozen pretrained backbones, which limits the ability to fully assess how scaling interacts with fairness when models are allowed to adapt more deeply to downstream tasks. While linear probing isolates representation quality from fine-tuning effects, it may underestimate the potential of larger models to mitigate bias through end-to-end adaptation, especially in domains like medical imaging where domain-specific features are critical. This design choice restricts the generalizability of the findings to scenarios involving full fine-tuning or task-specific architectural modifications.

2. The work evaluates fairness primarily through AUROC disparity ratios across predefined demographic groups, which, while common, may overlook other important fairness notions such as equalized odds, demographic parity, or calibration across subgroups. Moreover, the binary treatment of gender (male/female) fails to account for non-binary or transgender identities, reflecting a limitation inherent in the datasets but not critically interrogated by the authors. This narrow framing of protected attributes could mask more nuanced forms of bias, especially in intersectional contexts where multiple identities interact.

3. All models are pretrained exclusively on ImageNet-1k, a natural-image dataset with known biases and limited relevance to medical domains. While this setup helps control for pretraining confounders, it also means the observed fairness behaviors may not generalize to models pretrained on more diverse or domain-aligned data (e.g., large-scale medical image corpora). The study’s conclusions about the insufficiency of scaling might be specific to this particular pretraining regime and may not hold for foundation models trained on broader, more representative datasets.

4. The fairness loss introduced in the second set of experiments is applied only during the downstream linear probing phase, not during pretraining. This raises questions about whether the observed plateau in fairness gains is a fundamental limit of scaling or simply a consequence of applying fairness constraints too late in the pipeline. The paper acknowledges that fairness should be incorporated earlier but does not empirically test this hypothesis, leaving a gap between its recommendations and demonstrated evidence.

**Questions:**

Please refer to the weaknesses.

---

> ### Author Response · Authors · 2025-11-21
>
> We thank the reviewer for recognizing our systematic study of scaling effects on fairness in vision models and for appreciating our detailed analysis across protected attributes and data domains.
>
> > W1.
>
> We acknowledge the review's valid concerns regarding the limitations of using linear probing with frozen backbones. We chose to focus our analysis on the backbone representation because scaling fairness in pretrained methods is the most challenging problem. During finetuning, we already have a task-specific dataset and metadata that could be used to enhance fairness. However, we agree that the limitation of not including finetuning in our analysis could provide a broader discussion on the ease of adaptation and raw performance [1].
>
> > W2
>
> We have extended our evaluation to include widely adopted fairness metrics such as Equal Opportunity and Demographic Parity.
>
> > W3.
>
> We acknowledge the reviewer with a valid point regarding the exclusive use of ImageNet-1k pretrained models and the associated limitations due to its natural image domain and known biases. Our choice to include both natural and medical imaging datasets aimed to address these concerns by examining fairness behaviors across domains. Notably, we demonstrate consistent fairness patterns in the CelebA dataset, which shares strong alignment with ImageNet-1k, suggesting that the observed phenomena are not isolated to a single domain. However, we recognize that the conclusions about scaling insufficiency may vary for foundation models pretrained on broader and more domain-diverse datasets, which is an important direction for future research to validate and extend our findings.
>
> > W4.
>
> We applied the fairness loss during the downstream linear probing phase because relevant metadata, such as age and gender, are available at this stage and can be incorporated into the fairness objective. In contrast, most pretraining techniques are self-supervised and rely exclusively on raw data, making the integration of fairness losses that require attribute metadata impractical during pretraining. We recognize this limitation and have already recommended extending fairness-aware techniques to the pretraining phase as a promising direction for future research.
>
> References.
>
> [1] Q. Garrido, M. Assran, N. Ballas, A. Bardes, L. Najman, and Y. LeCun, “Learning and Leveraging World Models in Visual Representation Learning,” Mar. 01, 2024, arXiv: arXiv:2403.00504. doi: 10.48550/arXiv.2403.00504.

---

> > ### Comment · Area_Chair_vWnU · 2025-11-28
> >
> > Dear Reviewer,
> >
> > Please make sure you read the authors' response and engage with them in the discussion before the end of the discussion period on **Dec 03 '25 09:00 PM UTC**. This is a hard deadline.
> >
> > Thank you for supporting quality peer review at ICLR.
> >
> > AC

---

### Official Review · Reviewer_iLNE · 2025-10-31

**Soundness:** 2
**Presentation:** 2
**Contribution:** 1
**Rating:** 2
**Confidence:** 5

**Summary:**

This paper investigates how scaling vision models and datasets affects algorithmic fairness, particularly in medical and natural image domains. Using Hierarchical Vision Transformers of varying sizes and datasets including CelebA, HAM10000, and CheXpert, the authors systematically examine fairness disparities measured by AUROC gaps across protected groups such as age and gender. They find that while scaling improves fairness for some attributes, especially gender, it does not universally mitigate bias, and fairness gains plateau beyond a critical scale even as overall accuracy continues to improve. Incorporating a fairness loss into the objective helps reduce disparities but also exhibits diminishing returns, highlighting that scaling alone is insufficient to achieve equitable performance. The results emphasize the need for fairness-aware objectives during pre-training to address structural and representational biases, particularly for localized attributes like age in medical imaging contexts.

**Strengths:**

* Strong motivation and relevance: The paper addresses an important and timely question — whether scaling up vision models and datasets inherently improves fairness. It challenges the common assumption that “bigger models are fairer,” providing a necessary empirical correction to this belief.
* Systematic empirical analysis: Scaling laws have rarely been examined from a fairness perspective in vision models, especially in medical imaging.
* Cross-domain evaluation: By comparing both natural and medical imaging domains (CelebA, HAM10000, CheXpert), the paper reveals that fairness behaviors are dataset- and domain-dependent.

**Weaknesses:**

* Limited coverage of experimental scope: The analysis is confined to linear probing with frozen backbones, which cannot capture fairness trends that may emerge under fine-tuning. Many modern downstream pipelines (e.g., LoRA, full fine-tuning) could exhibit different scaling–fairness dynamics, so the conclusions drawn from fixed-feature experiments are incomplete.
* Narrow exploration of fairness mechanisms: The study evaluates only one fairness-regularized loss. Given the variety of fairness interventions (e.g., adversarial debiasing, distributionally robust optimization, reweighting), it is unclear whether the reported “fairness plateau” is a general phenomenon or specific to this formulation.
* Weak connection to prior work on bias transfer: Although related studies have analyzed how bias propagates during pre-training and transfer (e.g., Lee et al., Continual Learning in the Presence of Spurious Correlations), this paper lacks a thorough discussion situating its findings within that literature. The related work section should explicitly link its motivation to those prior theoretical and empirical results.
* Scaling range too small for reliable laws: The changes in both model size (from 27M to 671M parameters) and dataset size (10–25 examples per class) are insufficient to meaningfully fit or verify scaling laws, which typically require orders-of-magnitude variation. The limited scale weakens the claim that observed effects represent “scaling laws.”

**Questions:**

* The authors claim that fairness gains plateau with scaling, suggesting the need for fairness-aware pretraining. However, since all experiments use frozen representations, how would the observed fairness–scaling relationship change under finetuning-based adaptation (e.g., LoRA or full finetuning)?

---

> ### Author Response · Authors · 2025-11-21
>
> We appreciate the review's thoughtful recognition of the paper’s strong motivation, systematic empirical design, and comprehensive cross-domain evaluation, which we believe are central to advancing understanding of fairness under scaling in vision models.
>
> > W1.
>
> We acknowledge the review's valid concerns regarding the limitations of using linear probing with frozen backbones. We chose to focus our analysis on the backbone representation because scaling fairness in pretrained methods is the most challenging problem. During finetuning, we already have a task-specific dataset and metadata that could be used to enhance fairness. However, we agree that the limitation of not including finetuning in our analysis could provide a broader discussion on the ease of adaptation and raw performance [1].
>
> > W2.
>
> This loss was chosen because it goes directly to the focus of our argumentation, penalizing group-level fairness violations. We will add other techniques to mitigate fairness that go in other directions to improve the discussion.
>
> > W3.
>
> We acknowledge the review suggestion and recognize the importance of connecting our findings to prior work on bias transfer, such as Lee et al.’s study on continual learning with spurious correlations. Due to time constraints, we will not include a review of that literature.
>
> > W4.
>
> We acknowledge the review concern regarding the parameter and dataset size ranges used to fit scaling laws. Vision models typically operate at smaller parameter scales compared to language models, making a range from 27 million to 671 million parameters a substantial variation. For instance, the Dinov2 family spans approximately 21 million to 1 billion parameters, while scaling to multi-billion parameter models like Dinov3 (up to 7 billion parameters) is a very recent development. Thus, our parameter range captures a significant portion of the current vision model spectrum.
>
> A limitation of our work is that we only evaluated using a linear probing protocol, for which 10–25 examples per class is a reasonable sample size [2]. We plan to extend our study with full fine-tuning using a larger dataset of approximately 200,000 samples to better explore scaling behaviors at larger data scales.
>
> > Q.
>
> Given that masked autoencoder (MAE) give your potential in a full finetuning this process could help the model to have better results in fairness, as show in natural language the finetuning process could better achieve new scaling laws [3], given that in the full finetuning we update the weights in the model using supervised learning in a dataset specific task and could utilize demographic data provided in the datasets. However, the pretraining stage remains the primary challenge for fairness optimization due to a lack of accessible metadata and limited techniques for incorporating fairness constraints in self-supervised learning.
>
> References.
>
> [1] Q. Garrido, M. Assran, N. Ballas, A. Bardes, L. Najman, and Y. LeCun, “Learning and Leveraging World Models in Visual Representation Learning,” Mar. 01, 2024, arXiv: arXiv:2403.00504. doi: 10.48550/arXiv.2403.00504.
>
> [2] X. Zhai, A. Kolesnikov, N. Houlsby, and L. Beyer, “Scaling Vision Transformers,” June 20, 2022, arXiv: arXiv:2106.04560. doi: 10.48550/arXiv.2106.04560.
>
> [3] B. Isik, N. Ponomareva, H. Hazimeh, D. Paparas, S. Vassilvitskii, and S. Koyejo, “Scaling Laws for Downstream Task Performance in Machine Translation,” Feb. 20, 2025, arXiv: arXiv:2402.04177. doi: 10.48550/arXiv.2402.04177.

---

> > ### Comment · Reviewer_iLNE · 2025-11-24
> > **Official comment**
> >
> > Thank you for the reply. While appreciate the effort to address my concerns, my concerns have not been fully addressed. So, I will keep my original rating.

---

### Official Review · Reviewer_RgjE · 2025-11-02

**Soundness:** 2
**Presentation:** 2
**Contribution:** 1
**Rating:** 2
**Confidence:** 4

**Summary:**

Paper evaluates 5 models on 3 binary tasks, looking at fairness as measured by ROCAUC ratio between groups and overall performance.

They explore adding a fairness loss that is never defined (eq. 3), which "penalizes the p-norm (*what is value is p?*) of the violation vector (*which definition are you using for this?*)".

The conclusion reached is that fairness is somewhat improved by scaling, but only up to a point with no improvement seen beyond that, and that this point differs from task to task.

**Strengths:**

The conclusion is likely true, but some familiarity with the fairness literature and existing papers, particularly "Why Is My Classifier Discriminatory?" by Irene Chen, Fredrik D. Johansson, David Sontag 2018 could provide some theoretical analysis of what's going on.

The writing is to a good standard.

**Weaknesses:**

There's two major concerns here.

1. The practical experiments are needlessly minimal. Each of the three datasets comes with a wide range of binary labels that are likely to have different fairness properties, and because the approach fixes the backbone and only updates a linear head, a model for every label could have been easily trained for roughly the same computational budget. It's very likely that different labels will have different fairness properties. Particularly on celbA, there's large amounts of label noise for the label "earrings" on men, but not on women, and this will lead to fairness concerns that can not be resolved by better model generalization.
2. There's no meaningful analysis of the results that tries to show what's going on. This is where looking at the literature, particularly works such as: "Why Is My Classifier Discriminatory?" and coming up with additional experiments/analysis to identify possible causes of the discrimination would go a long way.

Beyond this, the use of auc ratio as a fairness measure is non-standard and can mask a wide range of unfairness. For example, arbitrary equal opportunity or demographic parity violations can occur when two groups have the same AUC curve. Moreover, AUC is pretty unhelpful in deciding if a classifier works well when the label distribution is unbalanced (high AUC can mask a scenario where no threshold results in a classifier with acceptable recall and precision).

**Questions:**

What are you actually optimizing in the fairness loss? How was this chosen over all of the standard losses in the literature?

Can you justify the use of AUC ratio for fairness?

---

> ### Author Response · Authors · 2025-11-21
>
> I appreciate that you considered the conclusions valid and pointed out a good paper that goes in the same direction and brings an interesting discussion.
>
> > W1.
>
> We sincerely thank the reviewer for their insightful feedback and the recommendation to incorporate Equal Opportunity and Demographic Parity metrics. In response, we have included these fairness metrics in Table 1 of the revised manuscript. Notably, the results indicate that scaling the model improves Equal Opportunity from 0.18 to 0.54 and Demographic Parity from 0.16 to 0.50 in a well-aligned dataset, CelebA, demonstrating that model scaling can enhance fairness. Nonetheless, we emphasize that scaling alone is insufficient to fully address fairness concerns.
>
> > W2.
>
> We appreciate the reviewer’s suggestion to investigate the underlying causes of discrimination, as analyzed in works such as “Why Is My Classifier Discriminatory?”. Our work aligns with the direction of that paper, and we will include it. We show that scale is important for improving fairness. However, we propose that the limits of scaling are influenced by pretrained models primarily optimized for utility. Similar to the argumentation of the paper “Why Is My Classifier Discriminatory?”, “identifying clusters or subpopulations with high predictive disparity would allow for more targeted ways to reduce discrimination.” by Irene Chen, Fredrik D. Johansson, David Sontag 2018. Given that scale is important to fairness, we need techniques that do not rely on metadata to reduce discrimination in target groups.
>
> > Q.
>
> The fairness loss we optimize targets the overall disparity in model predictions across sensitive groups by minimizing the norm of the violation vector, which corresponds to differences in fairness-related statistics, such as positive rates, that we used. This loss was selected for direct focus on penalizing group-level fairness violations and could be used jointly with other losses like binary cross-entropy.
>
> We employ the Area Under the Curve (AUC) metric consistent with the original datasets used (CheXpert, HAM10000, CelebA) for performance evaluation. Recognizing limitations in fairness evaluation metrics, we have supplemented our analysis with Equal Opportunity and Demographic Parity measures to provide more comprehensive fairness assessment.

---

> > ### Comment · Reviewer_RgjE · 2025-11-25
> >
> > Thanks for your comments.
> >
> > I'm not likely to change my view on the text, so don't feel obliged to respond, but there are a few issues brought up by the rebuttal that you might want to think about in future revisions.
> >
> > Re: *W1*.
> >
> > > the results indicate that scaling the model improves Equal Opportunity from 0.18 to 0.54 and Demographic Parity from 0.16 to 0.50
> >
> > This is a bit confusing. Normally, for demographic parity and equal opportunity, lower is better. It looks like you're using non-standard measures.
> >
> > Re: *W2*
> >
> > > Similar to the argumentation of the paper “Why Is My Classifier Discriminatory?”, “identifying clusters or subpopulations with high predictive disparity would allow for more targeted ways to reduce discrimination.”
> >
> > That's not really what the paper argues. The point is that prediction error can be decomposed into label noise, bias, and variance. How it relates to your paper is: noise can't be improved by changing the model, while model scale/dataset size can have some effect on variance. So any fairness concerns driven by a difference in model noise between groups can not be resolved by altering model scale or dataset size.
> >
> > This kind of analysis might allow you to better explain the behaviour found in your experiments.
> >
> > >The fairness loss we optimize targets the overall disparity in model predictions across sensitive groups by minimizing the norm of the violation vector, which corresponds to differences in fairness-related statistics, such as positive rates, that we used.
> >
> > You really need to use an equation here.
> >
> > 1. It's still not clear which norm you're using. Is this the L_2 norm or L_1?
> > 2. Positive rate is a piecewise constant function with respect to the model parameters. You can't optimise this directly with gradient descent. You must be using some approximation of it.

---

> > > ### Author Response · Authors · 2025-11-28
> > >
> > > Considering the circumstances, I will not be submitting a rebuttal.
> > >
> > > > Re: W1
> > >
> > > Sorry for the confusion. I'm using the demographic parity/equal opportunity ratio, so a value of 1 means that all groups have the same selection rate.
> > >
> > > > Re: W2
> > >
> > > Thank you, this paper provides valuable insight into the behavior observed in our experiments.
> > >
> > > We will include the equation and explain it more clearly in the text.
> > >
> > > We use the fairness regularizer as defined in Eq. (8) of [1] and the violation vector is defined in Eq. (9) [1]
> > >
> > > 1. We use the L_1 of the violation vector.
> > > 2. We use an approximation, following the discussion in Appendix B.3 of [1]
> > >
> > > [1] M. Buyl, M. Defrance, and T. D. Bie, “fairret: a Framework for Differentiable Fairness Regularization Terms,” Apr. 10, 2024, *arXiv*: arXiv:2310.17256. doi: [10.48550/arXiv.2310.17256](https://doi.org/10.48550/arXiv.2310.17256).

---

### Meta-Review · Area_Chair_2onF · 2025-12-17

**Summary:**

While the reviewers agree the paper investigates the timely problem of whether scaling up vision models and datasets inherently improves fairness, there are also significant concerns on the fairness metrics, technical details, experiments, analysis, and related work.

**Reviewer Concerns:**

While the authors responded to the reviews, the reviewer concerns on the fairness metrics, technical details, analysis, and related work remain, and the proposed revisions are extensive.

**Reviewer Scores:**

Overall, I do not see a score change.
* Reviewer RgjE: based on the response to the rebuttal, I do not think the reviewer would have increased the score.
* Reviewer iLNE explicitly stated that the score will remain the same.
* Reviewer 5tEA: while there is no response to the author rebuttal, the concerns are similar to those of the other reviewers.
* Reviewer iS83 explicitly stated that the score will remain the same.

---

### Decision · Program_Chairs · 2026-01-26

Reject